# Towards Video Text Visual Question Answering: Benchmark and Baseline

**Minyi Zhao**[1][*] **Bingjia Li**[1][*] **Jie Wang**[2]**, Wanqing Li**[2]**, Wenjing Zhou**[2]**, Lan Zhang**[2]
**Shijie Xuyang**[1]**, Zhihang Yu**[1]**, Xinkun Yu**[1]**, Guangze Li**[1]**, Aobotao Dai**[1]**, Shuigeng Zhou**[1][†]
[1]Shanghai Key Lab of Intelligent Information Processing, and School of
Computer Science, Fudan University, Shanghai 200438, China
[2]ByteDance, China
{zhaomy20, bjli20, abdai20, sgzhou}@fudan.edu.cn
{wangjie.bernard, liwanqing.0415, zhouwenjing.233, zhanglan.11}@bytedance.com
{shijiexuyang, gzlifd}@gmail.com    {zhyu21, xkyu21}@m.fudan.edu.cn

## Abstract

There are already some *text-based visual question answering* (TextVQA) benchmarks for developing machine's ability to answer questions based on texts in images in recent years. However, models developed on these benchmarks cannot work effectively in many real-life scenarios (e.g. traffic monitoring, shopping ads and e-learning videos) where temporal reasoning ability is required. To this end, we propose a new task named *Video Text Visual Question Answering* (ViteVQA in short) that aims at answering questions by spatiotemporally reasoning texts and visual information in a given video. In particular, on the one hand, we build the first ViteVQA benchmark dataset named M4-ViteVQA — the abbreviation of **M**ulti-category **M**ulti-frame **M**ulti-resolution **M**ulti-modal benchmark for **ViteVQA**, which contains 7,620 video clips of 9 categories (i.e., *shopping*, *traveling*, *driving*, *vlog*, *sport*, *advertisement*, *movie*, *game* and *talking*) and 3 kinds of resolutions (i.e., 720p, 1080p and 1176×664), and 25,123 question-answer pairs. On the other hand, we develop a baseline method named T5-ViteVQA for the ViteVQA task. T5-ViteVQA consists of five transformers. It first extracts optical character recognition (OCR) tokens, question features, and video representations via two OCR transformers, one language transformer and one video-language transformer, respectively. Then, a multimodal fusion transformer and an answer generation module are applied to fuse multimodal information and generate the final prediction. Extensive experiments on M4-ViteVQA demonstrate the superiority of T5-ViteVQA over the existing approaches of TextVQA and VQA tasks. The ViteVQA benchmark is available in https://github.com/bytedance/VTVQA.

## 1 Introduction

Several datasets [1, 2, 3, 4, 5, 6, 7] have been built to facilitate the development of *visual question answering* (VQA) [1] methods and systems, but none of them consider the high-level scene text information that is ubiquitous in real-life scenarios and urgently needed for visually-impaired users [8]. Thus, researchers later proposed some *text-based VQA* (TextVQA) benchmarks [8, 9] to promote the research on jointly understanding both visual information and scene texts in images.

Though existing TextVQA benchmarks have greatly advanced TextVQA techniques, all these benchmarks focus on single well-photographed images, which makes the developed TextVQA models

---

[*]This work was mainly done while the first two authors are interns in ByteDance with equal contribution.
[†]Corresponding author.

36th Conference on Neural Information Processing Systems (NeurIPS 2022) Track on Datasets and Benchmarks.

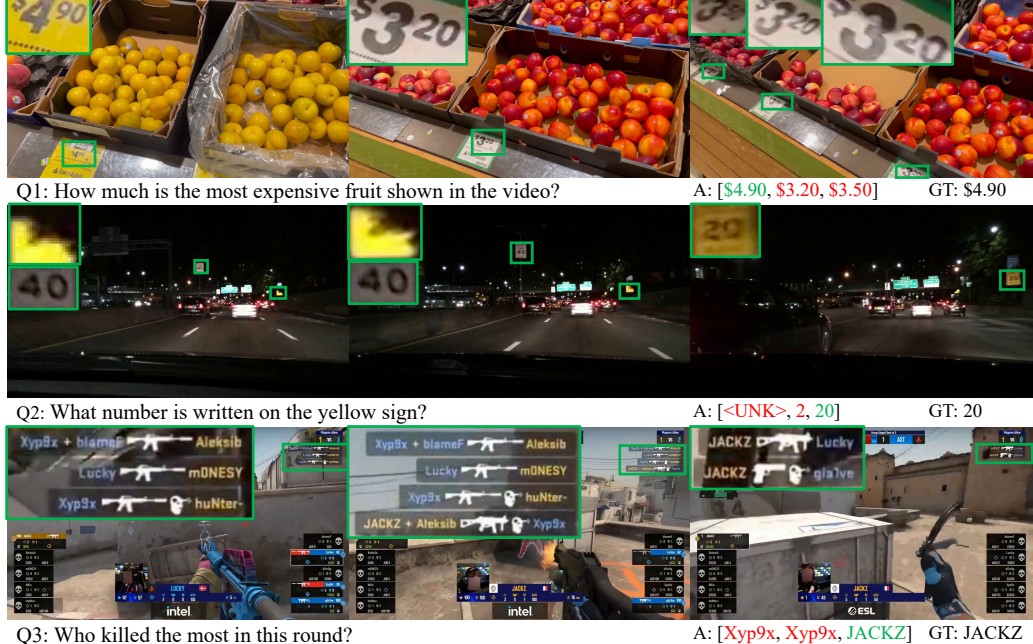

Q1: How much is the most expensive fruit shown in the video?    A: [$4.90, $3.20, $3.50]    GT: $4.90

Q2: What number is written on the yellow sign?    A: [<UNK>, 2, 20]    GT: 20

Q3: Who killed the most in this round?    A: [Xyp9x, Xyp9x, JACKZ]    GT: JACKZ

Figure 1: Some examples from our M4-ViteVQA dataset. 'A' indicates the answers returned by TextVQA models and wrong answers are colored in red. Leveraging temporal, textual, and visual information in the video is the only way to correctly answer the questions.

unable to answer questions related to or depending on consecutive video frames or events. This is a critical weakness of existing TextVQA methods, which has already been realized by some existing work [10], and seriously limits their applications. To make it simpler, let us see the examples in Fig. 1. The 1st case in Fig. 1 is querying the highest fruit price. TextVQA models will get three answers, *i.e.,* $4.90, $3.20, and $3.50 from three single frames, while the correct answer is $4.90. For the 2nd case in Fig 1, considering the scene texts in the video usually suffer from low quality due to motion blur and low resolution, which makes the TextVQA models prone to wrong answers because of lacking global understanding of the texts. As for the 3rd case, to get the correct answer the models must be able to infer temporarily. Unfortunately, such mechanism is unavailable in existing TextVQA models.

To solve the aforementioned problems, in this paper we propose a novel task named ***Video Text Visual Question Answering*** (ViteVQA in short), which aims at answering questions by jointly reasoning textual and visual information in a given video. For better understanding, let us go back to Fig. 1. To accurately answer the given questions, the models are required to leverage not only the semantics of texts (in the 1st case, $4.90 is the highest price), visual information (in the 2nd case, the models should recognize the yellow sign) and the spatial relationships among texts and objects (in the 3rd case, the models should know who kills who), but also the temporal relationships among different frames or events (in the 2nd and 3rd cases, the correct answer can be only obtained from the last frame). As ViteVQA is an extension to the TextVQA task, it is more general and has wider applications. Meanwhile, ViteVQA is also a more challenging task as it must jointly exploit both textual and visual information as well as temporal logic among video frames or events.

To support ViteVQA research, on the one hand, we build the first ViteVQA benchmark dataset, which is named **M**ulti-category **M**ulti-frame **M**ulti-resolution **M**ulti-modal benchmark for ViteVQA (M4-ViteVQA in short). M4-ViteVQA consists of 7,620 video clips of nine categories (i.e., *shopping*, *traveling*, *driving*, *vlog*, *sport*, *advertisement*, *movie*, *game* and *talking*) and three kinds of resolutions (i.e., 720p, 1080p and 1176×664), and 25,123 question-answer pairs (QA pairs). On the other hand, we develop a baseline method for the task, which is a novel model called T5-ViteVQA, as it consists of five transformers to conduct both textual and visual understanding as well as temporal reasoning over three modalities: texts from the video, a given question and a video. Specifically, T5-ViteVQA

first extracts optical character recognition (OCR) tokens in the form of temporal representation, question features, and video features via two OCR transformers, one language transformer and one video-language transformer, respectively. Then, a multimodal fusion transformer is employed to fuse and enhance these features. Finally, an answer generation module is applied to infer the answer from the OCR tokens and a given vocabulary.

Contributions of this paper are as follows: 1) We propose a novel task of *video text visual question answering* (ViteVQA), which is an extension to the TextVQA task and has broader applications. 2) To support ViteVQA research, we build the first high-diversity benchmark dataset M4-ViteVQA. 3) We develop a baseline method T5-ViteVQA for ViteVQA, which consists of five transformers to conduct joint reasoning over three modal inputs. 4) We conduct extensive experiments on M4-ViteVQA, which show that T5-ViteVQA outperforms the existing methods of TextVQA and VQA tasks.

## 2 Related Work

### 2.1 Text-based VQA

*Text-based visual question answering* (TextVQA) [8] is gaining popularity to answer text-related questions by reading and understanding scene texts in images. As a pioneering work, Singh *et al.* [8] proposed the first dataset TextVQA along with a new framework LoRRA that extends the VQA model Pythia [11] with an OCR attention branch. Following that, several other datasets were introduced. Biten *et al.* [9] built the dataset ST-VQA of daily natural scenes where questions can only be answered with texts in images. OCR-VQA [12] utilizes an existing dataset [13] of cover images of books, and contains around 200K QA pairs. DocVQA [14] focuses on the understanding of texts in documents. STE-VQA [15] is the first bilingual dataset containing both English and Chinese question-answer pairs. It also provides a bounding box for each question to indicate the area that contains the answer. Nevertheless, all these datasets focus on single static images while many real-world scenarios provide consecutive videos. Models trained on these TextVQA datasets cannot work well in the video scenarios where temporal reasoning ability is required.

### 2.2 Video Question Answering

As an extension of traditional VQA, *video question answering* (VideoQA) aims to answer questions about *video content*, requiring models to have spatiotemporal reasoning ability. There are already a number of datasets [16, 17, 18, 19, 20, 21, 22] for this task, which contain video clips of different scenes, while all the questions focus on the *visual content* of the videos. There are some other VideoQA datasets [23, 24, 25, 26, 27], most of which are about movies or TV series and provide additional texts like subtitles to help understand the videos. However, all these texts are explicitly provided in textual format, and questions in these datasets still focus on *visual content* while the texts just play an auxiliary role. Differently, ViteVQA considers all texts appearing in the scenes of videos, which can be single words or phrases in any possible fonts or orientations, thus cannot be recognized and understood easily. Furthermore, ViteVQA pays more attention to the interaction between texts and visual information in the videos.

### 2.3 Feature Representations in TextVQA and VideoQA Models

Most existing methods [28, 29, 30, 31, 32] for TextVQA utilize an OCR system [33] to detect and recognize texts in images and an object detector [34] to extract object region proposals. Then, the two modalities along with the question are reasoned jointly via a multimodal fusion transformer. Apparently, such a paradigm does not exploit temporal information in videos and our M4-ViteVQA dataset. While in VideoQA [35, 36, 37, 38, 39, 40], video features are obtained by sampling certain frames or extracted directly via a 3-D backbone [41, 42]. Some works [43, 44, 45, 46, 40, 47] use pre-training to obtain more comprehensive feature representations. In this paper, we extend all the representations of different modalities to the video level.

### 2.4 Video OCR Systems

Recently, the detection, tracking and recognition of texts in videos [48] have made great progress thanks to several video text spotting benchmarks [49, 50, 51, 52, 53] and models [54, 55, 56, 57, 58,

59, 60]. However, as mentioned in [53], there still remains a blank for the downstream application of texts in videos. This work is the first to propose and address the ViteVQA task, which is expected to broaden the research of TextVQA to videos and promote the understanding of texts in videos.

## 3 Benchmark

In this section, we start by describing how we collect and label the videos, and create QA pairs in M4-ViteVQA. Then, we present the statistics and analysis results of M4-ViteVQA, and compare our dataset with some related datasets to highlight the uniqueness, difficulty and diversity of the M4-ViteVQA dataset. Finally, we introduce the tasks and evaluation protocol of the benchmark. Documents on license, responsibility agreement, and accessibility are given in the supplementary materials.

### 3.1 Data Collection and Annotation

**Data collection.** To obtain abundant videos with various text types, we first selected nine different text-rich scenarios, which correspond to nine video categories: *shopping*, *traveling*, *driving*, *vlog*, *sport*, *advertisement*, *movie*, *game*, and *talking*. Then, 6 workers were employed to search qualified videos with texts from YouTube [3]. For the driving category, we collected additional videos from [51] to enrich the dataset. To avoid copyright violation, workers were required to try their best to download only videos that are available on YouTube with a Creative Commons CC-BY (v3.0) License. After collecting 1,150 raw videos, we further cropped

Table 1: The numbers of videos, frames and questions in each category of M4-ViteVQA.

| Category | #Videos | #Frames | #Questions |
|---|---|---|---|
| shopping | 847 | 155,275 | 3,892 |
| traveling | 1,154 | 219,880 | 4,291 |
| driving | 1,316 | 148,040 | 3,272 |
| vlog | 947 | 168,715 | 2,897 |
| sport | 665 | 133,979 | 2,072 |
| advertisement | 623 | 113,108 | 1,264 |
| movie | 719 | 103,429 | 1,449 |
| game | 709 | 155,645 | 3,672 |
| talking | 640 | 119,321 | 2,314 |
| Total | 7,620 | 1,317,392 | 25,123 |

these videos into shorter clips to discard frames without texts and shorten the lengths of videos. Then, we masked all private information in the videos, such as faces. Finally, it took about 30 days for the 6 workers to collect 8,511 video clips with texts.

**Data annotation.** In this stage, 11 native English-speaking workers were employed by crowdsourcing to create question-answer (QA) pairs. Similar to the annotation phase in [9], the process of designing QA pairs consists of two steps. In the first step, workers were required to come up with closed-ended questions that can be unambiguously answered by reasoning the texts in the corresponding video. The workers were asked to design three to seven QA pairs for each video. As a question may have different answers, so the workers can list multiple answers for each question, usually a complete answer plus a simplified one. Additionally, the workers were asked to attach each question two extra labels. The first label is from {"easy", "hard"} to indicate the difficulty of question answering. An "easy" question can be answered by first watching the whole video then picking key information from one single frame (*e.g.* the 3rd frame of the 2nd case in Fig. 1), while a "hard" question can be answered only by leveraging two or more frames (*e.g.* the 2nd and 3rd frames of the 3rd case in Fig. 1). It is worth mentioning that the so-called "easy" questions in our dataset are not really easy, they are just relatively easier than the so-called "hard" questions to answer. Actually, it is still challenging for TextVQA models to answer these "easy" questions (See Tab. 5 for more details). Concretely, temporal reasoning is still required to answer these questions.

The second label is from {"text", "vision", "knowledge"} to indicate what kind of information is required to answer the question. Concretely, a "text" question can be answered by purely understanding the semantics of texts in the video (*e.g.* to answer the question of the 1st case in Fig. 1 by outputting the highest price). A "vision" question is answered by jointly considering both the semantics of texts and the visual features of texts (e.g. color and layout) or the video features (e.g. objects and actions) (*e.g.* the 2nd case in Fig. 1). And a "knowledge" question can only be answered by exploiting external knowledge. The workers were encouraged to design "hard" and "vision" questions to increase the difficulty of the benchmark. After this step, we obtained 31,915 QA pairs. Then, we conducted a second step or the verification step, for which 8 additional workers (different from the 11 annotators)

---

[3]https://www.youtube.com/

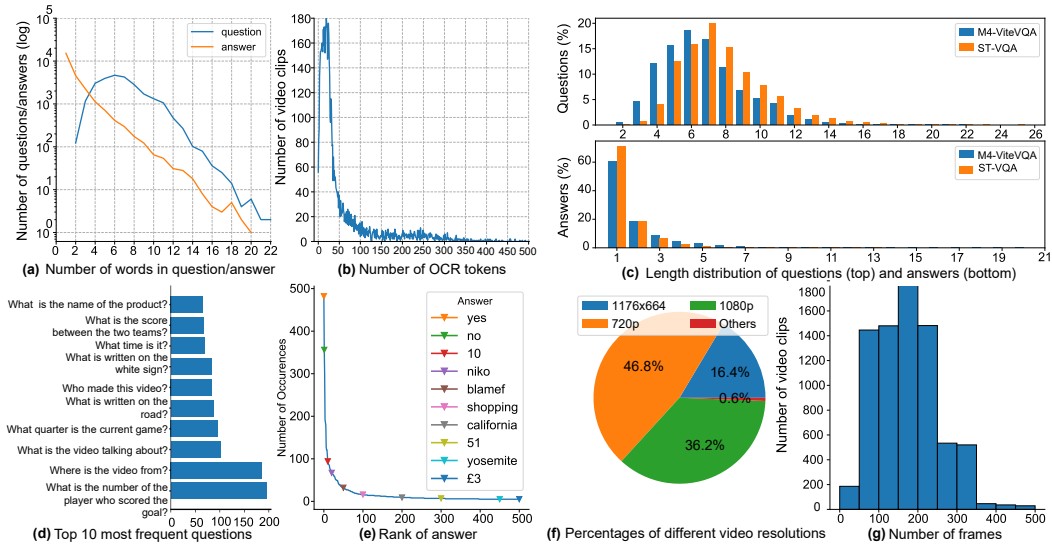

Figure 2: The statistics of questions, answers, OCR tokens and videos of M4-ViteVQA.

were employed to check the previously designed questions. Different from [9], the 8 workers have the right to delete text-unrelated questions to guarantee the quality of the dataset. As for some ambiguous questions that may generate inconsistent or contradictory answers, the authors decided whether or not to correct or delete them. The entire data annotation stage took nearly 50 days. Finally, we obtained 25,123 QA pairs from 7,620 different videos. The statistics of the dataset is given in Tab. 1.

## 3.2 Statistic, Analysis Results and Comparisons

We first analyze the numbers of questions, answers and OCR tokens in our dataset. Fig. 2(a) shows the question and answer distributions w.r.t. their lengths. The lengths of the majority (99%) questions and answers are under 14 and 10, respectively. Tab. 2 presents the average lengths of questions and answers of two existing TextVQA datasets and our dataset. As can be seen from Tab. 2, the average lengths of questions and answers of M4-ViteVQA are 6.75 and 1.94 respectively,

Table 2: Average lengths of questions and answers of TextVQA [8], STVQA [9], and our M4-ViteVQA

| Dataset | Question | Answer |
|---|---|---|
| TextVQA [8] | 7.05 | 1.63 |
| STVQA [9] | 7.79 | 1.55 |
| M4-ViteVQA | 6.75 | 1.94 |

they are quite similar to that of [8] and [9]. Concretely, the average length of questions of our dataset is a little shorter than that of [8] and [9], but the average length of answers of our dataset is longer than that of [8] and [9]. The distribution of the number of OCR tokens (extracted by methods in [53, 61]) is given in Fig. 2(b). As can be seen, most videos have 1 to 100 OCR tokens. The average number of OCR tokens in M4-ViteVQA is 56.92. We also compare the distributions of questions and answers with that of STVQA [9]. As can be seen in Fig. 2(c), the distributions of the two datasets basically follow the same law: the length of the questions first rises and then falls, while the length of the answers shows a downward trend.

Then, we present the statistics of the most frequent questions and the total occurrences of the 500 most common answers in our dataset in Fig. 2(d) and Fig. 2(e), respectively. We can see that there are common questions (*e.g.* "what is the video talking about?"), answers (*e.g.* numbers, shopping) and category-specific QA pairs (*e.g.* names of players and products). In what follows, we analyze the formats of questions and the unique numbers of answers in our dataset to highlight the diversity of M4-ViteVQA. The sunburst for the first 4 words in questions is given in Fig. 3. We can see that M4-ViteVQA contains diverse question types that cover various scenes. Concretely, we count the number of different formats of the first two words of the questions in our dataset, the result is 1,482, a bit larger than that of STVQA [9] (1,468), indicating that our dataset has more different question formats than [9]. Besides, the distributions of the two types of labels are: 0.87/0.13 for {"easy", "hard"} and 0.44/0.54/0.02 for {"text", "vision", "knowledge"}. As for the questions, the

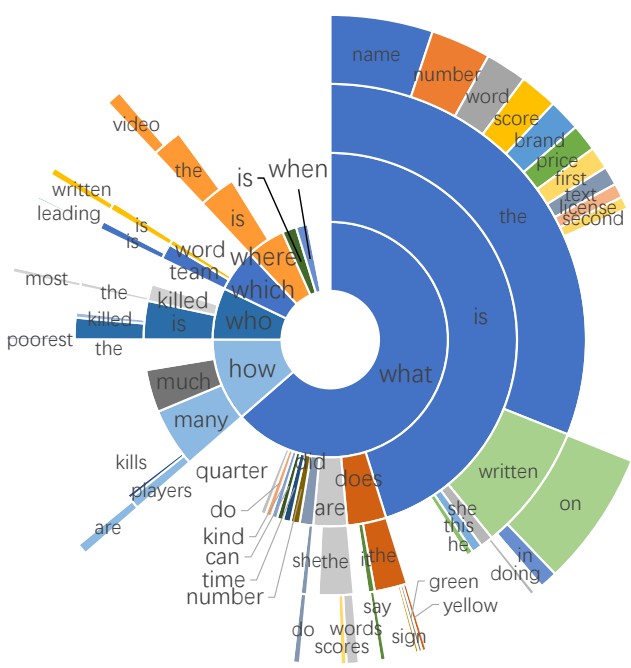

Figure 3: Distribution of the first four words in questions in M4-ViteVQA. Most questions start with "what".

Table 3: The numbers of videos (#V) and questions (#Q) in the three settings.

| Setting | Task1Split1 | | Task1Split2 | | Task2 | |
|---|---|---|---|---|---|---|
| | #V | #Q | #V | #Q | #V | #Q |
| Train | 5,373 | 17,869 | 5,444 | 18,220 | 4,772 | 14,855 |
| Val | 596 | 1,971 | 525 | 1,620 | 179 | 762 |
| Test | 971 | 3,183 | 680 | 2,103 | 290 | 1,221 |
| Extra | - | - | - | - | 1,018 | 4,223 |

percentage of "yes" and "no" in our dataset is very small (only 3.3%) and our dataset has 14,871 different answers among all the 26,293 answers. This means 56.5% of our answers are unique, which is roughly similar to that in [8] (57.6%) and [9] (60.2%). Considering the percentage of unique answers and the fact that our average answer length is 0.4 longer than that of the other datasets, we can say that the answers in our dataset are diverse.

Last, we present two statistics from the perspective of the videos to better demonstrate the diversity of the dataset: 1) The distribution of videos of different resolutions in M4-ViteVQA is shown in Fig. 2(f). There are three resolutions in our dataset: 720p (1280×720), 1080p (1920×1080) and 1176×664. 2) The statistics of the number of frames in the videos is illustrated in Fig. 2(g). The majority of the videos consist of 50 to 250 frames. However, some videos require the models to reason over 300 or even more frames. The different resolutions and lengths of videos not only enrich the diversity of the dataset, but also impose strict requirements on models' reading and temporal reasoning abilities.

### 3.3 Tasks and Evaluation Protocol

We define 2 tasks with 3 settings for the ViteVQA problem, namely "regular QA task" (Task1) and "domain adaption task" (Task2). In Task1, the model is trained and tested on all the nine categories of M4-ViteVQA, which is a regular setting. In order to meet the different requirements for the robustness of the model, we consider two data splits for Task1. The first one is called *Task1Split1* that is divided according to the 7,620 cropped videos, the second one is called *Task1Split2* that is divided by the 1,150 raw videos. Task1Split2 is more challenging than Task1Split1 since the content of videos of the same category may be quite different (*e.g.* various shopping venues and sports). In Task2, the model is trained with seven categories while tested on the remaining two categories. Task2 requires the model to deal with unlearned content and completely different category-specific questions, which is

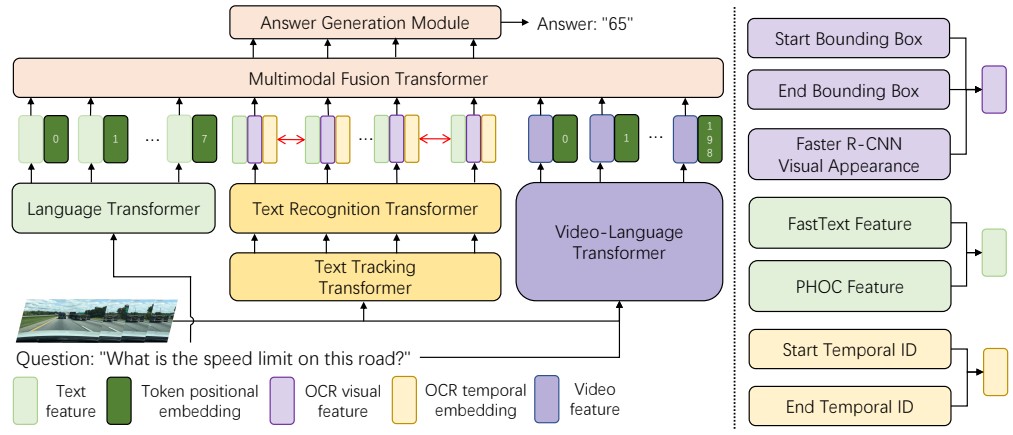

Figure 4: The architecture of T5-ViteVQA.

very challenging. The statistics of the three settings is given in Tab. 3. It is worth mentioning that for Task2, we provide an extra set, which can be used in different ways (*e.g.* semi-supervised learning, weakly supervised learning etc.) to improve the adaption ability of the model.

Two metrics are used in this benchmark to evaluate model performance. The first is *accuracy* used in existing TextVQA benchmarks [8, 9]. The second is the *average normalized Levenshtein similarity* (ANLS) [62, 9]. Comparing with the widely used accuracy, ANLS is more tolerant to recognition error, which is more suitable for ViteVQA due to the challenge of reading texts in videos.

## 4  Baseline

### 4.1  Architecture

Fig. 4 is the architecture of our baseline T5-ViteVQA, which mainly consists of five transformers and one answer generation module. Given a sample with a video $V$ and a question $Q$, we first extract the features of the question via a language transformer, the OCR token features with a text tracking transformer and a text recognition transformer, and the video features through a video-language transformer. Then, we adopt a multimodal fusion transformer to fuse these features. Finally, an answer generation module is stacked at the end to generate the answer from the OCR tokens and a given vocabulary. In what follows, we first describe the extraction of multimodal features, then introduce how to fuse them to answer the question, and finally present the training scheme.

### 4.2  Feature Extraction

**Question features.** Let $Q = \{Q_i\}_{i=1}^{L_q}$ be the sequence of the tokenized question tokens, where $L_q$ is the sequence length of the question, we embed these words into a sequence of $d$-dimensional feature vectors $X^q = \{x_i^q\}_{i=1}^{K}$ with a pretrained language transformer $T_l$ (*i.e.,* $X^q = T_l(Q)$ where $X^q \in \mathbb{R}^{K \times d}$). In T5-ViteVQA, $T_l$ is implemented by a BERT [63] model.

**OCR features.** Given the video $V$ with $L_v$ frames, we first employ a text tracking transformer $T_t$ to obtain bounding boxes and tracking ids of the texts in the video frame by frame. After that, we read the texts according to their bounding boxes via a text recognition transformer $T_r$. Then, for the OCR results that have the same tracking *id*, we merge all the bounding boxes, temporal *ids*, and the recognition results to obtain a temporal OCR token representation. Suppose there are $L_o$ predicted OCR tokens, the $i$-th OCR representation $O_i$ can be written as follows: $O_i = \{\{x_{i,j}^{bbox}\}_{j=1}^{L_t^i}, x_i^{s\_tid}, x_i^{e\_tid}, \{x_{i,j}^{ocr\_text}\}_{j=1}^{L_t^i}\}$ where $L_t^i \in [1, L_v]$ is the temporal length of the OCR token, $x_{i,j}^{bbox} \in \mathbb{R}^4$ is the bounding box of the $i$-th OCR token at time $j$, $x_i^{s\_tid} \in [1, L_v]$ and $x_i^{e\_tid} \in [1, L_v]$ record when the OCR token appears and disappears (Ergo, we have $L_t^i = x_i^{e\_tid} - x_i^{s\_tid} + 1$), and $x_{i,j}^{ocr\_text} \in \mathbb{R}^{L_{i,j}^{ocr} \times |\mathcal{A}|}$ is the recognition result of the $i$-th OCR token at time $j$ ($L_{i,j}^{ocr}$ denotes the length of the OCR and $|\mathcal{A}|$ is the size of the alphabet).

After obtaining a set of $L_o$ OCR tokens in a video through an external OCR system, as shown in Fig. 4, for the $i$-th token in the $L_o$ OCR tokens, we further reduce the redundancy in $O_i$ by extracting 1) the start bounding box $x_{i,1}^{bbox}$ and the end bounding box $x_{i,L_o^i}^{bbox}$ as the representative visual geometric features; 2) a 2048-dimensional visual appearance feature $x_i^{frcn}$ from a Faster R-CNN [34] detector via RoI-Pooling the first bounding box of the OCR token whose recognition result occurs the most times (denote the index of this token by $k$); 3) a 300-dimensional FastText [64] embedding of the recognition result of $x_{i,k}^{ocr\_text}$ to provide essential sub-word information, namely $x_i^{ft}$; 4) a 604-dimensional Pyramidal Histogram of Characters (PHOC) [65] vector $x_i^{phoc}$ to represent characters presented in the token, which is more robust to OCR error [28]; 5) the start temporal *id* $x_i^{s\_tid}$ and the end temporal *id* $x_i^{e\_tid}$ to provide crucial information for temporal reasoning. It is worth mentioning that for the $L_o$ OCR tokens, we first sort them according to the reading order and then use a two-layer transformer (red arrows in Fig. 4) to build their contextual dependence for the enhancement of their semantic features $x_{i,k}^{ocr\_text}$. After rescaling the bounding boxes by dividing the width or height of the video, we project or embed each feature into $d$-dimensional space, and sum them up [28] (after layer normalization) to get the final representation of the OCR feature:

$$x_i^{ocr} = LN(W_1 x_i^{ft} + W_2 x_i^{frcn} + W_3 x_i^{phoc} + E_1([x_i^{s\_tid}, x_i^{e\_tid}])) + LN(W_4[x_{i,1}^{bbox}, x_{i,L_o^i}^{bbox}]), \quad (1)$$

where $W_1, W_2, W_3$, and $W_4$ are learnable projection matrices, $E_1$ is a embedding layer, $LN(\cdot)$ is layer normalization, and $[\cdot, \cdot]$ denotes the concatenate operation. In T5-ViteVQA, $T_t$ and $T_r$ are implemented by TransVTSpotter [53] and ABINet [61], respectively.

**Video features.** We apply a video-language transformer $T_{vl}$ to extract question-guided visual information to aid the reasoning. Specifically, we uniformly sample $L_{vl}$ frames from the video $V$ and combine them with the question to extract the video features. Let the sampled $L_{vl}$ frames be $V_{vl}$. The final video representation is defined as $X^v = T_{vl}(V_{vl}, Q)$ where $X^v \in \mathbb{R}^{L_{vf} \times d}$ and $L_{vf}$ is the length of the video features. In our paper, we use All-in-one [40] to implement $T_{vl}$.

### 4.3 Feature Fusion and Answer Generation

Given the three multimodal features $X^q$, $X^{ocr} = \{x_i^{ocr}\}_{i=1}^{L_o}$ and $X^v$, we first employ a multimodal fusion transformer $T_f$ that consists of $K$ transformer layers [66] to enhance these features in multimodal context via self-attention mechanism. Then, the $L_o$ enhanced $d$-dimensional OCR tokens are fed into an answer generation module [28, 44] to infer the answer by selecting words from the OCR tokens in the video and a given vocabulary that is obtained by merging all the tokens in the training set in our experiments (see Sec. 5.1).

### 4.4 Training Scheme

Following [28, 44], we use teacher-forcing technique [67] and multi-label binary cross-entropy loss $\mathcal{L}_{bce}$ to train the model. Let $y_{pred}$ be the predicted result processed by sigmoid function and $y_{gt}$ be the ground truth, the loss of T5-ViteVQA can be written as follows:

$$\mathcal{L}_{bce} = -y_{gt}\log(y_{pred}) - (1 - y_{gt})\log(1 - y_{pred}). \quad (2)$$

## 5 Performance Evaluation

### 5.1 Implementation Details

T5-ViteVQA is implemented in PyTorch1.8. All experiments are conducted on 8 NVIDIA Tesla V100 GPUs with 32GB memory and the same random seed 13. The model is trained using AdamW [68] optimizer with a learning rate of $10^{-4}$. The batch size is set to 64. The warm-up learning ratio and warm-up iteration are as 0.2 and 1,000. $L_q, L_o, L_{vf}, L_{vl}, d$, and $K$ are set to 20, 200, 198, 3, 768 and 4, respectively. The selected frames $V_{vl}$ are resized to $224 \times 224$ for saving computational resource. The vocabulary is obtained by merging all the tokens in the training set.

Table 4: Performance comparison on the M4-ViteVQA dataset.

| Method | Task1 | | | | | | | | Task2 | | | |
|---|---|---|---|---|---|---|---|---|---|---|---|---|
| | Split1 | | | | Split2 | | | | 7 others → *shopping,talking* | | | |
| | Val | | Test | | Val | | Test | | Val | | Test | |
| | Acc | ANLS | Acc | ANLS | Acc | ANLS | Acc | ANLS | Acc | ANLS | Acc | ANLS |
| Random | 0.56 | 0.021 | 0.60 | 0.025 | 1.54 | 0.030 | 1.38 | 0.034 | 1.57 | 0.030 | 0.41 | 0.023 |
| All "yes" | 1.67 | 0.018 | 1.51 | 0.016 | 1.85 | 0.019 | 1.81 | 0.019 | 3.41 | 0.036 | 2.62 | 0.026 |
| All "no" | 0.96 | 0.012 | 1.32 | 0.016 | 1.60 | 0.021 | 1.66 | 0.022 | 2.89 | 0.029 | 2.87 | 0.030 |
| Biggest OCR box | 2.59 | 0.054 | 2.67 | 0.060 | 3.21 | 0.077 | 3.02 | 0.053 | 2.38 | 0.056 | 2.38 | 0.056 |
| Most frequent OCR | 3.86 | 0.060 | 3.80 | 0.056 | 6.17 | 0.089 | 4.61 | 0.068 | 5.12 | 0.070 | 3.93 | 0.059 |
| Upper bound | 68.44 | 0.710 | 68.05 | 0.714 | 67.22 | 0.702 | 64.76 | 0.677 | 66.40 | 0.698 | 62.49 | 0.646 |
| Human | 78.08 | 0.825 | 85.27 | 0.893 | 75.98 | 0.832 | 78.41 | 0.828 | 83.33 | 0.859 | 82.26 | 0.851 |
| JuskAsk | 10.81 | 0.154 | 10.05 | 0.141 | 7.16 | 0.100 | 5.47 | 0.086 | 4.86 | 0.067 | 3.60 | 0.067 |
| All-in-one-B | 11.47 | 0.153 | 10.87 | 0.148 | 6.85 | 0.092 | 5.66 | 0.078 | 4.20 | 0.050 | 3.28 | 0.046 |
| M4C | 18.66 | 0.242 | 17.91 | 0.238 | 13.58 | 0.172 | 11.36 | 0.166 | 9.16 | 0.128 | 7.52 | 0.125 |
| T5-ViteVQA | **23.17** | **0.301** | **22.17** | **0.291** | **17.59** | **0.231** | **16.68** | **0.238** | **12.30** | **0.161** | **9.29** | **0.136** |

Table 5: Detailed performance comparison between M4C and T5-ViteVQA on the validation set of Task1Split1. We do not present the results on the knowledge set because its sample number is too small in the validation set.

| Set | M4C | T5-ViteVQA |
|---|---|---|
| Easy | 19.30 | **25.09** |
| Hard | 9.02 | **14.26** |
| Text | 17.26 | **23.08** |
| Vision | 18.36 | **24.21** |
| Total | 17.91 | **23.17** |

Table 6: Ablation study on the features of T5-ViteVQA. The metric is the accuracy on the validation set of Task1Split1.

| $X^{ocr}$ | | | $X^v$ | Val Acc. |
|---|---|---|---|---|
| Text | Visual | Temporal | | |
| ✓ | ✓ | ✓ | ✓ | **23.17** |
| ✗ | ✗ | ✗ | ✓ | 11.51 |
| ✓ | ✗ | ✓ | ✓ | 20.79 |
| ✓ | ✓ | ✗ | ✓ | 19.78 |
| ✓ | ✓ | ✓ | ✗ | 22.36 |

## 5.2 Experimental Results

We start by giving the performance upper bound and lower bound of ViteVQA as well as human evaluation on the benchmark, then present the performance comparison between our baseline and existing methods of related tasks. Finally, we introduce the results of ablation study.

**Performance upper/lower bound and human evaluation.** Here, we present the performance upper and lower bounds as well as human evaluation results on the benchmark. All results are given in Tab. 4. The lower bound values (denoted as "Random", "All yes", "All no", "Biggest OCR box", "Most frequent OCR" in Tab. 4) are obtained by randomly picking OCRs from the videos, always outputting "yes", always outputting "no", picking OCRs with the biggest boxes, and picking OCRs with the most frequent occurrences, respectively, and the upper bound values (denoted as "Upper bound" in Tab. 4) are achieved by correctly picking texts, which can be used to evaluate the performance of the OCR system used in the ViteVQA task. Obviously, all the lower bounds perform poorly on the M4-ViteVQA dataset, which demonstrates the difficulty of the dataset. Besides, we can see that the upper bounds are lower than human evaluation results (indicated by "Human" in Tab. 4), which demonstrates the difficulty of reading texts in videos. Besides, the accuracy of human varies from 75.98% to 85.27%, which is very close to the human evaluation result (84.01%) in TextVQA [8]. This also indicates that human can handle ViteVQA well.

**Comparison with existing methods.** We reimplement three recent models of related tasks for performance comparison, including M4C [28] for TextVQA, JustAsk [39] for VideoQA and All-in-one [40] for video-language pretraining. The experimental results are given in Tab. 4.

Obviously, JuskAsk and All-in-one perform undesirably in the ViteVQA task since they cannot read texts in videos. M4C performs better than JuskAsk and All-in-one, but is clearly inferior to our method in all three settings because it cannot do temporal reasoning in videos. Although T5-ViteVQA outperforms the existing techniques in all three settings, as can be seen in the 3rd row and the last row in Tab. 4, there is still a huge gap between T5-ViteVQA and human evaluation. This indicates that ViteVQA is a difficult task for machine and worthy of further investigation. It is also worth mentioning that the difficulty of the three settings is increasing. All the models perform much worse in Task2 without applying any domain adaption techniques, while human can answer the questions in Task2 well. Therefore, how to use additional data to enhance the generalization power of the model is a significant issue to work on. In addition, we compare the performance of M4C and T5-ViteVQA

on the subsets introduced in Sec. 3.1. As can be seen in Tab. 5, our method T5-ViteVQA performs much better than M4C on these different subsets, which demonstrates the advantage of our method on the ViteVQA task because of its temporal reasoning ability. It is worth mentioning that although M4C [28] can correctly answer 40.6% questions in traditional TextVQA task [8], it performs poorly on the "easy" questions in the task of ViteVQA, only 19.3% are correctly answered and much worse on the "hard" questions. These results show the difficulty of the proposed M4-ViteVQA dataset.

**Ablation study.** We also conduct ablation study to check the design of T5-ViteVQA. As mentioned in Sec. 4, three modal features are extracted in our method, and for the OCR features, we extract textual, temporal and visual information to enrich the representation. In order to see how these features affect the performance of ViteVQA, we design some variants that ignore some specific features. The experimental results are given in Tab. 6. As can be seen in Tab. 6, the best performance is achieved when all the features are used (the 1st row). Besides, the variant that ignores all OCR features has the worst performance (the 2nd row), which explains the importance of OCR tokens in ViteVQA. The performance after dropping the video features $X^v$ is also deteriorated (the 5th row), but the degradation is not as much as that of ignoring visual (the 3rd row) and temporal (the 4th row) features of OCR. We notice that the recent TextVQA works [28, 45] also report this observation. This shows that the usage of video features (*i.e.*, the visual object features in TextVQA) should be improved in both TextVQA and ViteVQA tasks.

# 6   Limitations and Future Work

This paper has two limitations. On the one hand, M4-ViteVQA supports only question answering. On the other hand, T5-ViteVQA does not use pre-training or domain adaptation techniques to further boost the performance. Therefore, it is worthy for future work to extend M4-ViteVQA to text-based video captioning and retrieval tasks to enrich the video text understanding area. For the second limitation, pre-training or domain adaptation solutions are interesting and promising research topics.

In summary, as a new problem, ViteVQA opens a new direction for VQA or TextVQA over videos, and this work may inspire new research momentum to this area.

# 7   Conclusion

In this paper, we propose and address a novel problem called *video text visual question answering* (ViteVQA), which requires the model to answer a given question by reading texts and visual information from videos and do temporal reasoning over consecutive events or frames in videos. To support ViteVQA research as a novel problem, we curate the first ViteVQA benchmark dataset named M4-ViteVQA which consists of nine categories of videos with three different resolutions, 7,620 video clips and 25,123 question-answer pairs. We also develop a ViteVQA model as the baseline called T5-ViteVQA, which mainly consists of 5 transformers. T5-ViteVQA first extracts question features, OCR features and video features from three different modal inputs, then fuses these features to generate the final answer. Extensive experiments on M4-ViteVQA show the superiority of our method to the existing techniques of TextVQA, VQA and video-language pretraining.

## Acknowledgments and Disclosure of Funding

The work was supported in part by a ByteDance Research Collaboration Project. We thank the following people from ByteDance AI Data Service for labeling M4-ViteVQA: Siew Chin Low, Hailey Kwong, Muhammad Nazhan Naqib Bin Mohd Akib, Szvetlana Melnicsenko, Maria Elena Romo Espinoza, Marta Ramos Baonza, Fernando Pereira, Henrique Revez, Maria Pilar Iglesias Pedreira, Za-K, Weng Kin Law, Selvaraj A/L Alurasamy.

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
