# OpenReview forum: "Towards Video Text Visual Question Answering: Benchmark and Baseline"
_NeurIPS.cc/2022/Track/Datasets_and_Benchmarks — NeurIPS 2022 Datasets and Benchmarks _

### Official Review · Reviewer_JRq3 · 2022-07-27

**Rating:** 7
**Confidence:** 4
**Correctness:** I did not find any major correctness …

**Strengths:**

S1. A novel task combining several dimensions. Video + language + visual-text task is novel.

S2. Data collection is non-trivial. A sophisticated collection process is done and the community will always appreciate this kind of manual effort (in contrast to auto-generated dataset).

S3. Human performance is estimated, and a large gap is presented for the research community to close.

**Weaknesses:**

W1. Dataset is small, which results in several limitations.

Each category only contains 1k-4k questions in total (Table 1). After splitting into test set, each category would only contain few hundreds of examples. This limits the robustness of per-category accuracy evaluation due to too few samples. This might also results in the need of external feature extractor (Faster R-CNN, FastText, and PHOC), which leads to a much more complex design and more overhead.

W2. Writing sometimes over-hype the dataset

Videos are naturally multi-frame, although the final model only uses 3 frames and I'm not sure if it can be seen as video. Table 1 also counts the total frame to try make a large dataset impression, but not fully utilized. Multi-resolution seems to be trivial. Online videos comes in different resolution, and the model downscales to 224 x 224 uniformly (L277). Text-Video-QA is already multi-modal. Considering W1, the lack of per-category evaluation also makes the multi-category argument weak. In my opinion, the "M4" is an overclaim that should be fixed in paper revision.

W3. Baseline might be too weak. Experiment section needs to be improved.

Given the fact that the model only takes 224x224 resolution and 3 frames, it is not surprising that human can achieve much better accuracy given the full video. More experiments should probably be done in higher resolution and frame rate. If the task can simply be solved by this, it would probably not live long in the community. It is not clear from the writing whether the BERT and All-in-one transformer are pre-trained and finetuned. Also, it is not clear why these 2 are chosen. Perhaps authors can try different text models (GPT, T5) and video encoder (Swin transformer).

Overall, I still lean positive about this paper. However, authors should try to address my above concerns during rebuttal and perhaps give revision plan for camera ready.

---------------------------- after rebuttal -----------------------------

Thanks for addressing my above concerns. The experiments are now stronger.

**Additional Feedback:**

A1. L sometimes refers to sequence length (L_vl, L_v), and sometimes number of layers (L_f). K, N, M also denotes sequence lengths. Perhaps rewriting the notations can help readability.

A2. Why Bi-LSTM in OCR features? Wouldn't a transformer layer work better or be more consistent?

A3. How is human performance done? No detail is provided. Are human workers also limited to 224 x 224 resolution and 3 frame? Definition of "Upper bound" is also confusing (L286-288). How to use it to "evaluate performance of the OCR systme used"? (Sec 5.2)

A4. I would suggest some more lower bound accuracy estimations such as: all "yes", all "largest OCR box".

----------------------------- after rebuttal --------------------------

Thanks for the response. I hope my suggestions helped the paper.

**Clarity:**

The writing is mostly clear. However, some claims are over-hyped. Please see weaknesses.

**Documentation:**

Supplementary materials describe the detail and contain code. An external URL is provided to access the dataset.

**Ethics:**

I do not see serious ethics issues to the best of my knowledge.

**Relation To Prior Work:**

Related works are properly discussed.

**Summary And Contributions:**

This paper propose a new multi-modal dataset, namely M4-ViteVQA, to challenge the research community with a difficult Text + Video question answering task. Different from previous video-QA tasks, the model is expected to understand "visual text" in the scene, rather than text from human dialogue / narration. Moreover, the authors claim that temporal reasoning (over video frames) is also required to solve the task. A baseline model is proposed to establish Transformer-based models' performance on this task. However, there is still a large gap to human performance.

---------------- after rebuttal -----------------

The authors have addressed my concerns in the rebuttal. I have increased my final rating to 7. The contribution of the dataset itself is non-trivial; and it outweighs the weaknesses in my opinion.

---

> ### Author Response · Authors · 2022-08-10
> **Response to Reviewer JRq3 (1/2)**
>
> Thank you very much for your helpful comments and suggestions. In what follows, we try to address your concerns.
>
> Response to W1:
>
> The size of our dataset is a bit smaller than that of STVQA[9] which consists of 31k QA pairs, **but bigger than or similar to that of some VideoQA datasets** (e.g. MovieQA [27] (10k), KnowIT VQA [69] (24k), TVQA+ [25] (29k) …). But please note that our dataset is for a new task --- test based video QA. Labeling a text-based video dataset is a much labor-consuming task since the annotators have to watch and understand the texts and visual information in the video many times. We recruited **19 workers for this work** and costed them about **1,000 working days** in total to collect, label, verify, and check the dataset. Finally, this dataset will be an open source for the community, and we will continue to update and expand the dataset.
>
> Response to W2:
>
> “M4” is the name we used to describe the characteristics of the dataset. Maybe it is not the most proper name for the dataset, but it does describe the characteristics of the dataset: Multi-category Multi-frame Multi-resolution and  Multi-modality. Here, we want to emphasize that dataset is dataset, method is method. How the method handles the data does not impact the characteristics of the dataset. Anyway, we will consider your suggestion and rethink the name for the dataset.
>
> Response to W3:
>
> We have the following points:
>
> 1. BERT and All-in-one transformer are first pre-trained and then finetuned on our dataset.
>
> 2. We conducted experiments with higher resolution and more frames, the results  are as follows:
> | Resolution | Input frames | Val Acc | Val ANLS | Test Acc | Test ANLS |
> |--------------|-----------------|-----------|--------------|------------|---------------|
> | 224*224    | 3        | 22.07  | 0.282    | 20.23    | 0.266     |
> | 224*224    | 9        | 21.40  | 0.285    | 20.23    | 0.268     |
> | 336*336    | 3        | 21.65  | 0.284    | 20.67    | 0.267     |
>
> As can be seen above, higher resolution and frame rate can not solve the problem, which demonstrates the difficulty of the new task. Possible reasons are: recent TextVQA methods emphasized the significance of texts over visual objects [45] and relevant visual features are not sufficiently used by existing methods[70]. Such problems are further exacerbated in ViteVQA due to substantial temporal and spatial redundancy in videos. Therefore, redundant information irrelevant to the questions may burden the reasoning and the following answer generation. We think this new task can offer opportunities for some interesting research topics such as frame selection, text&question-aware video representation, and spatiotemporal redundancy reduction.
>
> 3. BERT is widely used in TextVQA models [28,29] to extract question features. Different text models like GPT and T5 can be used to generate the answer but need massive training data to develop. For example, LaTr [45] crawls 77 million documents from the website to train a T5 structured model, which is too high-consuming and unsuitable to serve as a baseline for ViteVQA. In our experiments, directly finetuning T5 to generate the answer suffer from out-of-vocabulary issue and finally gets a 5% accuracy decrease.
>
> 4. Actually, we have used Swin transformer as the video encoder, and the results are as follows:
> | Method         | Acc   | Params. (M) | Flops (G) |
> |------------------------|----------|---------------|------------|
> | All-in-one       | 22.07  | 33        | 16.9     |
> | Swin Transformer | 22.41  | 49        | 26.3     |
>
> Although Swintransformer can achieve 0.34 accuracy increase (1.5% improvement over All-in-one), it introduces 16M additional parameters (48.5% bigger than All-in-one) and 9.4G flops (55.6% bigger than All-in-one). To balance performance and computational cost, we do not use Swintransformer.
>
> 5. Note that the first TextVQA model LORRA[8] gets an accuracy of 26.56% and is very similar to that of T5-ViteVQA (22.07%). This shows that our baseline is not “too weak”.

---

> > ### Author Response · Authors · 2022-08-10
> > **Response to Reviewer JRq3 (2/2)**
> >
> > Response to A1:
> >
> > Thanks for pointing out the writing problems, we will fix these problems in the revised manuscript.
> >
> > Response to A2:
> >
> > Thanks for the suggestion! Replacing the 2-layer Bi-LSTM of T5-ViteVQA with 2 transformer layers does improve performance from 22.07/0.282 to 23.17/0.3008 on the validation set of task1split1 and from 20.23/0.266 to 22.17/0.2908 on the test set. We will update our method T5-ViteVQA in the revised version following your suggestion.
> >
> > Response to A3:
> >
> > Human performance is obtained by asking workers to answer the questions. Human workers can view the full video of the original resolution to give the answer.
> >
> > As for upper bound, in our dataset, all the questions should be answered via reading and then picking the texts appearing in the video plus "yes”&”no". Therefore, as long as the ground truth answer appeared in the texts read by the OCR system, the question will be treated as correctly answered in the definition of upper bound, namely correctly picking the answer (L286). Please note that in the ideal situation, the accuracy rate of the upper bound should be 100%, that is, the current OCR system succeeds in correctly reading all the texts in the video. Therefore, the upper bound can also be used to evaluate the performance of the OCR systems. A higher upper bound means the OCR system can read more accurate texts from the video.
> >
> > Response toA4:
> >
> > We tried four new lower bounds, including “all yes”, “all no”, “largest OCR box”, and “most frequent OCR”, the results are as follows:
> > |                   | Acc on T1S1 Devset | ANLS on T1S1 Devset | Acc on T1S1 Testset | ANLS on T1S1 Devset |
> > |:--------------------------:|:-------------------------:|:-----------------------------:|:------------------------:|:---------------------------:|
> > |     All 'Yes'      |        1.67        |        0.018       |         1.51     |        0.016      |
> > |      All 'No'     |        0.96        |        0.012        |         1.32     |        0.016      |
> > |  Largest OCR box |        2.89        |        0.054        |         2.67     |        0.060      |
> > | Most frequent OCR |        3.86        |        0.060        |         3.80     |        0.056      |
> >
> > As can be seen above, although the bound of “largest OCR box” and “most frequent OCR” outperform “random OCR”, they still perform poorly on our dataset. This demonstrates the difficulty of our dataset.
> >
> > References:
> >
> > [69] Garcia, Noa, et al. "KnowIT VQA: Answering knowledge-based questions about videos." Proceedings of the AAAI Conference on Artificial Intelligence. Vol. 34. No. 07. 2020.
> >
> > [70] Wang, Qingqing, et al. "Towards reasoning ability in scene text visual question answering." Proceedings of the 29th ACM International Conference on Multimedia. 2021.

---

> ### Author Response · Authors · 2022-08-25
> **Response to Reviewer JRq3**
>
> Dear Reviewer JRq3,
>
> We have properly addressed all your concerns and submitted a revised version. Could you please kindly re-evaluate our paper based on the current situation? If you have any further questions, we are also very glad to discuss them.
>
> Thanks,
>
> Authors

---

### Official Review · Reviewer_PKrW · 2022-07-28
**The paper present one novel task, video text visual question answer, create one dataset, and propose one baseline models. The task is interesting and inspiring. The proposed model is also interesting. What I am concerning is the creating of the database, especially the question and answers, which are very simple. As such, the studies on the database may not be able to push the boundaries of the multimodal learning.**

**Rating:** 6
**Confidence:** 5
**Clarity:** yes

**Strengths:**

1. The paper proposed one novel task, namely he video text visual question answering.
2. One dataset is created with annotated questions as well as answers.
3. One benchmark method is also proposed.

**Weaknesses:**

My main concerns is the create dataset, especially the collected questions as well as the annotated answers.
1. According to the statistic and analysis results in Page 5, the lengths of the majority questions and answers are under 14 and 10. And the average lengths of questions and answers are 6.75 and 1.94. It seems that the questions as demonstrated in Figure 2 is very simple with a very simple format, namely “waht is XXXX”, and the answers are also very simple ,such as “yes”, “no”, and so on. In such a situation, the vqa task becomes very simple, which only exploit the characteristics of the short questions and answers.
2. The expression of the questions and answers is not diversity, which make the studies conducted on the dataset may not be able to reveal the true characteristics of the dataset.

**Additional Feedback:**

no

**Correctness:**

It seems to me that the dataset is not constructed in one perfect way ,which still presents some weakness.

**Documentation:**

yes

**Relation To Prior Work:**

yes

**Summary And Contributions:**

1. The paper proposed one novel task, namely he video text visual question answering.
2. One dataset is created with annotated questions as well as answers.
3. One benchmark method is also proposed.

My main concerns is the create dataset, especially the collected questions as well as the annotated answers.
1. According to the statistic and analysis results in Page 5, the lengths of the majority questions and answers are under 14 and 10. And the average lengths of questions and answers are 6.75 and 1.94. It seems that the questions as demonstrated in Figure 2 is very simple with a very simple format, namely “waht is XXXX”, and the answers are also very simple ,such as “yes”, “no”, and so on. In such a situation, the vqa task becomes very simple, which only exploit the characteristics of the short questions and answers.
2. The expression of the questions and answers is not diversity, which make the studies conducted on the dataset may not be able to reveal the true characteristics of the dataset.

---

> ### Author Response · Authors · 2022-08-10
> **Response to Reviewer PKrW**
>
> Thank you for your comments. In what follows, we respond to your two major concerns:
>
> 1. About simplicity of the dataset.
>
> Actually, in our point of view, our dataset is **NOT simple**.
>
> Firstly, the lengths of questions and answers of our dataset are normal and acceptable. For your reference, we present the average lengths of questions and answers of our dataset and two existing text based VQA datasets TextVQA [8] and STVQA [9] as follows:
> |            | Question | Answer |
> |----------------- |------------|-----------|
> | TextVQA[8] | 7.05     | 1.63   |
> | STVQA[9]  | 7.79     | 1.55   |
> | Ours       | 6.75     | 1.94   |
>
> As can be seen above, the average lengths of questions and answers of our dataset are quite similar to that of [8] and [9]. Concretely, the averaged length of questions of our dataset is a little shorter than that of [8] and [9], but the averaged length of answers of our dataset is longer than that of [8] and [9]. So from the perspective of lengths of questions and answers, our dataset is NOT simpler than the existing datasets.
>
> Secondly, there are many unique question formats in our paper (Please notice that in Fig. 2(d, e), only the top frequent questions and answers are displayed). As can be seen in Fig. 3 of our paper, our questions have various formats, including but not limited to: What is, What does, What are, How many, How much, Who is, Which team, Where is, Is, When, and etc… Concretely, we count the number of different formats of the first two words of the questions in our dataset, the result is 1,482, a bit **BIGGER** than that of STVQA[9] (1,468), indicating that our dataset has more different question formats than [9]. These also show that the questions of our dataset are not simple.
>
> Thirdly, the answers are not simple. The percentage of ‘yes’ and ‘no’ in our dataset is very small (only 3.3%) and our dataset has 14,871 different answers among all the 26,293 answers. This means 56.5% of our answers are unique, which is roughly similar to that in [8] (57.6%) and [9] (60.2%). Considering the percentage of unique answers and the fact that our average answer length is 0.4 longer than that of the other datasets, the answers in our dataset are **NOT** simple.
>
> Last but not least, M4C[28] achieves an accuracy of 40.55% and 38.05% on [8] and [9], respectively, but only gets an accuracy of 18.66% on our dataset. This also indirectly shows that our dataset is NOT simple.
>
> 2. About the diversity of the dataset.
>
> As mentioned above, our dataset has more diverse question formats than STVQA (1,482 v.s. 1,468). Furthermore, as can be seen in question distribution of our dataset (see Fig. 3 in our paper) and that of STVQA (see Fig. 3 in [9], or click [here](https://github.com/bytedance/VTVQA/blob/master/Imgs/Question_distribution.png) for convenience), obviously, our dataset has more diverse questions like Is, When, and etc. Besides, our dataset has less questions starting with “What”. These demonstrate the diversity of questions in our dataset.
>
> As for answers, our dataset has 14,871 unique answers among all the 26,293 answers, which means 56.5% of our answers are unique, very similar to that in [8] (57.6%) and [9] (60.2%).
>
> In summary, considering both length distribution and diversity of questions and answers of our dataset, we can see that our dataset is absolutely NOT simple. The most important point lies in that this dataset is proposed for a new task ViteVQA, which is more challenging than the traditional TextVQA task.

---

> > ### Comment · Reviewer_PKrW · 2022-08-29
> > **The authors somewhat addressed my concerns, with some concerns still remaining.**
> >
> > Thank you very much for the rebuttal.
> >
> > As stated by your rebuttal, I agree with that the constructed dataset is not simple, comparing with the TextQA and STVQA in terms of not only the lengths of the questions and answers, but also the performances of M4C. However, the questions and answers in real life should be much more complicated, comparing with the authors’ constructed dataset.
> >
> > Although the authors somewhat addressed my concerns. I am still concerning whether the dataset can mimic the real question and answers in real life situation.

---

> > > ### Author Response · Authors · 2022-08-29
> > > **Response to Reviewer PKrW - Ⅱ**
> > >
> > > Dear Reviewer PKrW,
> > >
> > > Thank you very much for your prompt feedback. As for your concern on simulating real questions and answers, actually, this is our first concern. As you can see from the affiliations of the authors, this dataset is a collaborative work between industry and academia. From the start of the project, our aim is to build a dataset that can effectively support applications of video text VQA in real scenarios, which drives us to do our best to create questions and answers as close to real situations as possible. To this end, on the one hand, we collect nine real-life categories, i.e., shopping, traveling, driving, log, sport, advertisement, movie, game and talking, which make our constructed dataset closer to real-life situations than existing TextVQA datasets, for example, TextVQA [8] that collects images from Open Image, a classification dataset and OCR-VQA[12] that contains only cover images of books. On the other hand, different from previous datasets, we encourage our recruited workers (data annotators) to write complicated and hard questions (See Sec 3.1 of our paper). This also makes our dataset more diverse (e.g. question formats) than existing datasets. Finally, as this work is supported and participated by industry, we will continue to expand and update the dataset by introducing more videos of different scenarios and designing more complicated questions to cover more real-life situations in the future.

---

> ### Author Response · Authors · 2022-08-25
> **Response to Reviewer PKrW**
>
> Dear Reviewer PKrW,
>
> We have properly addressed all your concerns and submitted a revised version. Could you please kindly re-evaluate our paper based on the current situation? If you have any further questions, we are also very glad to discuss them.
>
> Thanks,
>
> Authors

---

### Official Review · Reviewer_p2Q1 · 2022-07-28
**A benchmark and baseline method for video text VQA.**

**Rating:** 5
**Confidence:** 4
**Correctness:** The claims are correct.
**Clarity:** The paper is well written.

**Strengths:**

1. The video text visual question answering task has never been explored before. This paper is the first one to explore this new task, collect datasets, and design methods for it. It might shed light on future research in this direction.
2. The proposed ViteVQA task is challenging because it requires spatial reasoning, temporal reasoning, reasoning on the text and visual contexts at the same time. None of the previous approaches is able to handle this task well.
3. The authors collect a dataset for this task and conduct in-depth analysis on the dataset statistics.

**Weaknesses:**

1. Although this dataset is for video text VQA, 87% of the questions can be answered with the information from a single frame (``easy'' questions), according to lines 179-180.
2. The authors designed a "regular QA task'' and a ``domain adaptation task'', but the proposed baseline method are only designed for regular QA task and it is not mentioned how to deal with the domain adaptation task.
3. The proposed approach framework for this task is a transformer-based model with common designs. The technical contribution of the proposed framework is not strong enough. Most of the designs and the transformers are from previous work. For example, $T_t$ and $T_r$ are implemented by TransVTSpotter [53] and ABINet [61], respectively. $T_{vl}$ is implemented by All-in-one [40]. The fusion transformer consists of the transformer layers [66] and the generation module borrows [28,44].

Post-rebuttal:

The authors addressed part of my concerns, but I still believe that the baseline should be stronger. Especially there should be a baseline for the domain adaptation task. Although there are no existing techniques to solve this problem, the authors should at least provide a simple baseline so that following works can compare with the benchmark.

**Additional Feedback:**

N/A

**Documentation:**

There is sufficient detail on the dataset information.

**Ethics:**

There are no ethical concerns.

**Relation To Prior Work:**

It is clearly discussed how this work differs from previous contributions.

**Summary And Contributions:**

This paper proposes a new task of Video Text Visual Question Answering (ViteVQA), which extends the previous text-based visual question answering task into the video domain. The proposed task aims at answering questions by reasoning texts and visual information spatiotemporally in a given video.  The authors build the first Multi-category Multi-frame Multi-resolution Multi-modal benchmark for ViteVQA. They also develop the first baseline method, T4-ViteVQA, for this task. It  first extracts optical character recognition (OCR) tokens, question features, and video representations, and then fuses multimodal information and generates the final predictions.

---

> ### Author Response · Authors · 2022-08-10
> **Response to Reviewer p2Q1**
>
> Thank you for your constructive comments. Below are our responses to your concerns:
>
> Response to W1: We want to point out that the so-called “easy” questions in our dataset are NOT really easy, they are just relatively easier than the so-called “hard” questions to answer. Actually, it is still challenging for textvqa models to answer these “easy” questions. Concretely, **temporal reasoning is still required** to answer these questions. For instance, give a video of shopping at a fruit stand and a question of “How much does lemon cost?”, to answer this question, the model must first find the frames that lemon appears, usually there are more than one frame having lemon, and then from those relevant frames, to find the correct answer. This is absolutely NOT an easy task. And unfortunately, existing textvqa models cannot answer such a question well. As can be seen in Tab. 4 of our paper, although M4C[28] can correctly answer 40.6% questions in traditional TextVQA[8] task, it performs poorly on the “easy” questions in the task of ViteVQA, only 19.3% are correctly answered.
>
> Response to W2: First, one contribution of the ViteVQA benchmark is the definition of domain adaptation, which has not considered in existing TextVQA benchmarks (e.g. TextVQA[8] and STVQA[9]). Therefore, there are no existing techniques can be used to solve this issue in ViteVQA. Second, as a submission for the datasets and benchmarks track, we focus on defining the task of ViteVQA and build the dataset. Methodology is left for future work. We hope this benchmark work can inspire the multimedia community’s research interest in the domain adaptation setting.
>
> Response to W3: Firstly, this is the first work that addresses the challenges of temporal feature representation and multimodal fusion to build a powerful and simple baseline for ViteVQA. Secondly, as a paper for the dataset and benchmark track, our focus is on the construction of dataset, rather than the technical contribution. Thirdly, recent methods also widely use common designs to do the task of TextVQA (e.g., LaTr[45] uses ViT, T5, and Amazon OCR). And considering the fact that the proposed T5-ViteVQA has the advantages of simplicity and easy-to-follow, our solution is effective enough to serve as a baseline for ViteVQA.

---

> > ### Author Response · Authors · 2022-08-29
> > **Further Response to Reviewer p2Q1**
> >
> > Dear Reviewer p2Q1,
> >
> > We sincerely thank you again for your time and effort in reviewing our paper. To further address your concerns, we additionally provide some responses:
> >
> > Further Response to W1:  In addition to demonstrating the difficulty of the so-called "easy" questions via the performance of M4C, we provide some visualization cases of some ''easy'' questions to better explain the difficulty of the "easy" questions. As can be checked in [here](https://github.com/bytedance/VTVQA/blob/master/Imgs/Fig-Rebuttal.pdf) (some cases already given in our supplementary materials), to answer these questions, models must read the whole video and then extract key information from the relevant frame, for example, the last frame in the top case and the first frame in the bottom case, to generate the correct answer. Obviously, random selection can not answer these so-called "easy" questions. These visualizations also demonstrate the difficulty of the "easy" questions.
> >
> > Further Response to W3: We have updated our baseline by introducing a new 2-layer transformer to enhance semantic features. Our new baseline lifts the performance from 22.07/0.282 to **23.17/0.3008** on the validation set of task1split1 and from 20.23/0.266 to **22.17/0.2908** on the test set.

---

> ### Author Response · Authors · 2022-08-25
> **Response to Reviewer p2Q1**
>
> Dear Reviewer p2Q1,
>
> We have properly addressed all your concerns and submitted a revised version. Could you please kindly re-evaluate our paper based on the current situation? If you have any further questions, we are also very glad to discuss them.
>
> Thanks,
>
> Authors

---

### Author Response · Authors · 2022-08-21
**Paper revision.**

Dear reviewers, we have revised our submission according to your concerns. Here, we summarize the main changes to our submission's latest revision.
1. We add comparisons with some related datasets to highlight the difficulty and diversity of our dataset in Sec. 3 to address the concerns of Reviewer p2Q1 and Reviewer PKrW.
2. We improve our baseline T5-ViteVQA by replacing the Bi-LSTM with a two-layer transformer according to the suggestion given by Reviewer JRq3.
3. We add new lower bounds to Tab. 4 to better demonstrate the difficulty of the benchmark.
4. We rewrite the notations to help readability as suggested by Reviewer JRq3.

---

### Meta-Review · Area_Chair_ryaw · 2022-09-08

**Recommendation:** Accept
**Confidence:** 4

**Metareview:**

The paper presents a new video text VQA dataset, along a new baseline. The reviewers’s key concerns are around the size and difficulty of the dataset; the authors successfully address those during the discussion and revision period. The dataset is likely to be of interest to the community and was non-trivial to construct.

---

### Decision · Program_Chairs · 2022-09-16

Accept